# The Evolution of Affordable Technologies in Liquid Biopsy Diagnostics: The Key to Clinical Implementation

**DOI:** 10.3390/cancers15225434

**Published:** 2023-11-16

**Authors:** George Alexandrou, Katerina-Theresa Mantikas, Rebecca Allsopp, Calista Adele Yapeter, Myesha Jahin, Taryn Melnick, Simak Ali, R. Charles Coombes, Christofer Toumazou, Jacqueline A. Shaw, Melpomeni Kalofonou

**Affiliations:** 1Centre For Bio-Inspired Technology, Department of Electrical & Electronic Engineering, Imperial College London, London SW7 2BT, UK; katerina-theresa.mantikas16@imperial.ac.uk (K.-T.M.); calista.yapeter18@imperial.ac.uk (C.A.Y.); m.jahin18@imperial.ac.uk (M.J.); taryn.melnick16@imperial.ac.uk (T.M.); c.toumazou@imperial.ac.uk (C.T.); 2Leicester Cancer Research Centre, Department of Genetics and Genome Biology, University of Leicester, Leicester LE2 7LX, UK; rca17@leicester.ac.uk (R.A.); js39@leicester.ac.uk (J.A.S.); 3Department of Surgery and Cancer, Imperial College London, London SW7 2AZ, UK; simak.ali@imperial.ac.uk (S.A.); c.coombes@imperial.ac.uk (R.C.C.)

**Keywords:** cancer diagnostics, liquid biopsy, personalised treatment, emerging technologies

## Abstract

**Simple Summary:**

This review aims to highlight the usage of circulating tumour DNA and circulating tumour cells in various manners for the care of cancer patients. The different technologies that are currently employed using these biomarkers are mentioned and contrasted with another whilst also discussing their limitations such as affordability and scalability. The review also aims to bring light to newer emerging technologies in the space of liquid biopsy that have yet to be approved by a regulatory board but have been developed with the notion of affordability and scalability in mind. These factors of technology are found to be important in order to provide cutting edge diagnostic and monitoring regimes as they can lead to personalised treatments and patient stratification for all.

**Abstract:**

Cancer remains a leading cause of death worldwide, despite many advances in diagnosis and treatment. Precision medicine has been a key area of focus, with research providing insights and progress in helping to lower cancer mortality through better patient stratification for therapies and more precise diagnostic techniques. However, unequal access to cancer care is still a global concern, with many patients having limited access to diagnostic tests and treatment regimens. Noninvasive liquid biopsy (LB) technology can determine tumour-specific molecular alterations in peripheral samples. This allows clinicians to infer knowledge at a DNA or cellular level, which can be used to screen individuals with high cancer risk, personalize treatments, monitor treatment response, and detect metastasis early. As scientific understanding of cancer pathology increases, LB technologies that utilize circulating tumour DNA (ctDNA) and circulating tumour cells (CTCs) have evolved over the course of research. These technologies incorporate tumour-specific markers into molecular testing platforms. For clinical translation and maximum patient benefit at a wider scale, the accuracy, accessibility, and affordability of LB tests need to be prioritized and compared with gold standard methodologies in current use. In this review, we highlight the range of technologies in LB diagnostics and discuss the future prospects of LB through the anticipated evolution of current technologies and the integration of emerging and novel ones. This could potentially allow a more cost-effective model of cancer care to be widely adopted.

## 1. Introduction

Cancer management remains a major challenge in healthcare, with an estimated 9.6 million cancer-related deaths worldwide and a projected rise of 14–22 million new cases every year [1]. Liquid biopsy (LB) technologies aim to detect cancers at the earliest possible stage to increase survival rates and offer patients personalized treatments for more efficient results. In 2013 and 2016, the first LB tests detecting CTCs (CellSearch^®^ CTC enumeration platform [2]) and ctDNA (cobas EGFR mutation kit v2 [3]), respectively, were FDA approved in the US. Since then, there have been a growing number of approved LB tests for clinical use. Despite these efforts, cancer mortality still proves to be the second-highest cause of death worldwide. The burden of cancer mortality is still incredibly relevant even with the rise of approved LBs. Socioeconomic factors are key in understanding cancer mortality. When investigating mortality rates by ethnicity and income, significant biases can be observed, some of which may result from underfunded healthcare systems, often found in minority communities, thereby leaving individuals without healthcare access and reducing their chances for detection and treatment options. Since the COVID-19 global pandemic had severe implications to healthcare systems and particularly cancer care providers, delays occurred across screening, diagnosis, monitoring, and access to treatments [4]. This has led to an increase in cancer mortality rates and is expected to result in elevated rates in the coming years, due to deferred cancer screening and routine diagnostic check-ups in the UK and the US [4]. In the US, the Centers for Disease Control and Prevention (CDC) reported four times the number of recent projected cancer deaths due to COVID-19 [5]. Studies and public data statistics have shown that there has been a drop between 86% and 94% in preventative cancer screenings for breast, cervical, and colon cancer, due to COVID-19 disruptions [6]. Specifically in the UK, recent findings have reported an increase in mortality rates of up to 10% particularly for breast cancer and 15% for colorectal cancer, based on 5-year survival rates [7]. Decreased hospital screening programs have played a significant role towards this outcome as well as delays across surgical interventions and treatment administration. This phenomenon exposes the importance of affordability and accessibility, emphasizing the need for future postpandemic healthcare models to prioritise improvements in diagnostic technologies.

This review will explore various technologies for testing molecular tumour-based biomarkers, including ctDNA and CTCs, with an emphasis on emerging LB technologies that are pushing the boundaries of traditional cancer diagnostics and clinical management. Primary focus will be on newer technologies that prioritise affordability and accessibility, whilst still enabling better patient stratification, improved diagnosis, and treatment switching, with a projection onto how the future of cancer diagnostics could be reshaped with the incorporation of innovative and cost-effective technologies.

## 2. Liquid Biopsies—Clinical Significance

Intratumour heterogeneity is a large factor for metastasis and recurrent disease and is one of the primary reasons that late-stage cancer becomes difficult to cure and manage, often leading to cancer mortality [8]. To fully assess cancer’s mutational composition to provide data for better detection methods, monitoring of metastatic markers, identifying treatment resistance, and personalising treatment more effectively, a biopsy is required [9]. Current tissue biopsy methods are limited to the collection of tumour tissue samples through surgical intervention. While tissue biopsies are still largely used to aid in the staging and grading of cancers, they present certain restrictions when relied upon as the only source of mutational information for a patient’s cancer. LBs are less invasive tests that simply require a blood sample, saliva, or urine to investigate the presence of tumour-specific biomarkers. Subsequently, an LB can be performed at regular intervals to dynamically monitor a patient’s progression, with existing research outcomes to have recognised the importance of blood as an abundant source of various analytes for LBs, such as cell-free nucleic acids and exosomal vesicles [10]. Repeated sample acquisition is an issue with tissue samples, further highlighting the benefits of an LB. Tumour biopsies provide real information on disease staging and provide a foundational mechanism for initial disease prognosis and assessment. LBs, however, while not intended to replace tumour biopsies, can allow a high-throughput molecular analysis of the tumour, looking into the crucial aspect of clonal evolution and cancer heterogeneity while offering clinicians a noninvasive way of monitoring genetic and epigenetic changes over the course of a disease [8].

Ideally, the tests following an LB should be informative, with high sensitivity and specificity for certain biomarkers. However, given the vast molecular knowledge derived from research and the abundance of detailed molecular biology techniques, such as sequencing and PCR mutational testing, the affordability of these tests should be considered. The blood markers discussed in this review, circulating tumour cells (CTCs) and circulating tumour DNA (ctDNA), are notable examples, though they are not the only blood biomarkers that can be utilised from liquid biopsy tests. CTCs are actively disseminated cancer cells [11], which can provide prognostic information on the patient, dependent on their abundance in circulation. ctDNA, on the other hand, derived as the tumour fraction of circulating free DNA (cfDNA) in the blood [12], can be of prognostic value based on its abundance, while also providing mutational and methylation status, which can inform clinicians on how a patient’s cancer pathology is evolving over the course of treatment or signify a mutational pathway that can be targeted via a specific therapy.

### 2.1. Circulating Tumour Cells (CTCs)

CTCs are cancerous cells found in the bloodstream that originate from the primary tumour or a metastatic site and are considered a precursor of metastasis [11]. CTCs are present in the bloodstream as single cells or in clusters with a short half-life of 25–30 min for single cells and 6–10 min for cell clusters [13]. Their stability is dependent on factors such as interaction with platelets, which protects them from lymphocytes and natural killer cells [14]. Not only are CTCs eliminated by the natural response of an immune reaction, but it has been reported that the majority of CTCs are eliminated by fluid shear stress. Only a very low proportion of CTCs survive both these mechanisms, and it is still not well understood how CTCs overcome shear flow [15]. Tumour cells become invasive through genetic changes, causing an epithelial–mesenchymal transition (EMT) [16]. The tumour cells that undergo EMT display “stem-cell-like” characteristics, increasing their metastatic potential and allowing them to disseminate into the blood, facilitating metastasis [17]. Monitoring CTC levels over a course of treatment could provide insight into how well the patient responds to the given treatment when CTC count drops, whereas an equivalent increase can indicate poor prognosis often linked to an aggressive or metastatic tumour [18,19].

### 2.2. Circulating Tumour DNA (ctDNA)

The concept that tumour somatic mutations could be identified from cell-free DNA (cfDNA) in cancer patients and found abundantly, compared with healthy individuals, was first recognised in 1977 by Leon et al. [20]. This has been further proven in modern scientific literature by studying the fraction of ctDNA and CTC count as a prognosis of the disease [12]. With DNA being a highly electrostatic molecule, it is often present in macromolecular structures or internalised in vesicles. These structures protect the DNA from circulatory responses that would otherwise denature the DNA. It is thought that each tumour type may have different mechanisms of DNA release. For instance, tumours such as colorectal cancer that have a high tumour cell loss factor (96%) clearly indicate that, due to high cell death rates, DNA is released through necrotic or apoptotic processes [18].

Other possible mechanisms for DNA release involve active secretion, where DNA is released in vesicles and can be taken up by other cells [21]. Moreover, ctDNA has a half-life of between 16 min and 2.5 h [9], giving a real-time snapshot of the tumour and its evolution, and therefore provides clinicians a more temporally accurate source of genetic information compared with solid tumour biopsies. Consequently, over the last decade, many technological platforms and tests have been developed to study ctDNA for many types and subtypes of cancer to help clinicians make treatment decisions and aid prognosis. ctDNA forms only a small component of cfDNA and is often highly fragmented, and its concentration is highly variable between cancer subtypes, which presents a challenge in molecular studies for liquid biopsies. As such, there are many technical challenges and limitations in many of the methods that currently exist.

## 3. DNA Detection Approaches for Liquid Biopsies

Numerous technologies have been used for research studies and commercially, but most approved methods are based on either traditional qPCR mutational testing, droplet digital PCR (ddPCR), or next-generation sequencing (NGS), as represented in Figure 1.

### 3.1. Traditional PCR-Based Tests

Mutational testing by qPCR has been used in liquid biopsy research studies, with the detection efficacy to rely on the design and optimisation of mutation-specific primers, with the under-study sample to then be analysed and the mutations’ frequencies to be quantified. CancerSEEK is a blood-based PCR analysis system of ctDNA that aims to detect cancer in early stages, with applications in breast, ovarian, lung, liver, colorectal, gastric, and oesophageal cancer [22]. The basis of this test is to detect mutations in 2001 genomic positions, across 16 genes, and assess the levels of eight cancer-associated protein biomarkers, such as HGF (hepatocyte growth factor) and PRL (prolactin) [22], with reported median sensitivity from 73% to 78% and a specificity of more than 99%: 7 out of 812 healthy controls scored positive. Sensitivities varied based on the cancer that was screened for; for instance, breast cancer only showed 33% sensitivity, but 70% sensitivity was depicted for five cancers: ovarian, liver, gastric, pancreas, and oesophageal, for which currently no routine screening procedures are available. The price has been suggested by the team to be less than USD 500 a test. In contrast, the Therascreen PCR kit by Qiagen detects 11 mutations on the individual *PIK3CA* gene from blood plasma ctDNA and is used to aid in treatment switching instead of early cancer screening. The PIK3 pathway is one the most prevalent and significant oncogenic signalling pathways in breast cancer, present in 70% of breast cancer cases [23]. The presence of mutations in this gene is associated with better patient response to the drug alpelisib (PIQRAY by Novartis), a therapy that aims to inhibit this PIK3 oncogenic pathway [24]. The Therascreen KRAS PCR kit detects 7 mutations (12 ALA, 12 ASP, 12 ARG, 12 CYS, 12 SER, 12 VAL, and 13 ASP) on the *KRAS* oncogene for metastatic colorectal cancer [25]. The Therascreen test kits cost approximately north of GBP 3000 but also require the use of a Rotor-Gene Q MDX instrument, which costs upwards of GBP 10,000. The Therascreen kits offer a specific use case to be valuable but still require an initial investment for a healthcare setting to perform these tests. Similarly, from both tissue and plasma samples, the Roche Diagnostics cobas v2 test detects 42 mutations in exons 18, 19, 20, and 21 of the *EFGR* gene, which codes for the tyrosine kinase receptor protein that is reportedly overexpressed in 60% of lung cancers [26,27]. By classifying patients with EGFR mutations, it can offer information to clinicians regarding personalised treatment options. For instance, EGFR tyrosine kinase inhibitors (EGFR-TKIs) are an effective therapy used in non-small cell lung cancer (NSCLC). Patients may develop EGFR tyrosine mutations that void this therapy altogether. The cobas EGFR mutation test v2 costs GBP 125 per test and requires the use of a standard real-time PCR instrument as opposed to Therascreen tests that also require their proprietary RGQ instrument [28].

### 3.2. Digital Droplet PCR

Digital droplet PCR (ddPCR), the next generation of quantitative PCR (qPCR), divides a sample into many partitions, whereby each partition contains one or two target molecules that undergo individual PCR reactions. This means that ddPCR has higher sensitivity than qPCR; where ddPCR can usually test a mutant abundance of down to 0.1%, qPCR obtains a frequency of approximately 10% [29]. However, ddPCR is more complex and has a higher risk for contamination due to multiple transfer and pipetting steps, while also relying on lengthy workflows executed by specialised scientists. BEAMing, a variant of ddPCR and commercially available as OncoBEAM, is the culmination of bead emulsification amplification and flow cytometry [30]. BEAMing has been a common research practice to study ctDNA because of its increased sensitivity of up to 0.02%, stemming from its ability to separate every DNA molecule in a sample into droplets, forming a massive parallel reaction [9]. OncoBEAM’s robustness and utility has been showcased for numerous cancer types, from the testing of PIK3CA status for metastatic breast cancer with a sensitivity of 81.6% to EGFR mutational status in NSCLC cancer with a sensitivity of 90% [31]. Results also indicated that OncoBEAM plasma results had low minimum allele frequencies with 0.2%, which is below the cobas EGFR mutation test [32]. The turnaround time for OncoBEAM is traditionally 2 days, which is longer than qPCR kits, whereby results are generated on the testing day, but much shorter than NGS, which typically requires a 1-week turnaround time. Overall, OncoBEAM and other ddPCR tests run higher costs than qPCR and require expensive instruments for generating diagnostic results but offer higher sensitivity than qPCR kits and are thus preferred for the analysis of multipanel testing and clinical validation studies [33].

### 3.3. Next-Generation Sequencing

Sequencing approaches vary from targeted efforts to whole exosomal sequencing (WES) or whole-genome sequencing (WGS) tests. Whilst ddPCR offers great sensitivity, the mutations to screen for must be known in advance, and only a handful of genetic alterations can be studied at a time. NGS is able to bypass this design parameter and screen numerous novel variants. The allele frequency of detection for most sequencing methods is 0.1% [9]. Popular companies that perform or offer sequencers for purchase are Illumina, PACBio, Qiagen, Thermo Fisher, and Agilent. NGS can be applied to targeted panels to increase sensitivity and reduce false positives. Such panels are TAM-Seq, Safe-Seq, and CAPP-Seq [9]. TAM-Seq can identify mutations as low as 2% with sensitivity and specificity over 97% [34]. To reduce error rates, Sysmex’s Safe-Seq utilises unique identifiers that are tagged on all DNA molecules that are analysed, essentially making it a highly specific and sensitive sequencing assay; it has an increased sensitivity and specificity of 98% [34]. Safe-Seq has recently been demonstrated for its use in early breast cancer diagnostics by looking at *TP53* and *PIK3CA* mutations [35]. GRAIL, a company dedicated to creating a platform for pan-cancer early stage screening, has recently begun a large-scale clinical trial using high coverage and breadth sequencing of patient plasma samples to try to create a large ctDNA database [36]. Some newer LB assays also come with companion devices, such as FoundationOne Liquid CDx, which was approved in the US by the FDA to detect over 300 genes with cfDNA using NGS. It has been recently approved to identify patients with BRAF V600E–mutated metastatic colorectal cancer [37]. Its companion device also reports blood tumour mutational burden and microsatellite instability as well as a wealth of other information that can aid in patient stratification and recommend approved therapies for patients [38].

Third-generation sequencing platforms, such as Oxford Nanopore Sequencing [39], combat the accessibility of traditional NGS sequencers by having a technology that can be easily miniaturised, such as the MinION. Nanopore sequencing works by recording electrical changes as DNA molecules pass through a protein nanopore. As each nucleotide passes through the nanopore, a characteristic voltage change is recorded that identifies which nucleotide has just passed through. Due to it working with low DNA input and its ability to sequence long reads of DNA, it has many benefits, such as forgoing PCR in its protocol steps. However, it is not designed for cell-free or fragmented small strands of DNA, such as with a liquid biopsy sample. It is not currently approved in this space; however, much research is still underway for applications in LB diagnostics, especially with copy number variations (CNVs) and methylation profiling.

When comparing costs of NGS tests with those of qPCR or ddPCR, prices vary greatly based on the platform. NGS involves instrument cost, price per run, and price for bioinformatic analysis and, as mentioned, has a larger turnaround time compared with other methods previously mentioned, typically around 1 week. However, the turnaround time for targetted panels is much less than WGS or WES comparatively. Some example prices for instruments are Illumina MiSeq, USD 128k; Ion torrent PGM, USD 80k; PacBio RS, USD 695K; and Illumina HiSeq 2000, USD 654k [40]. Therefore, it becomes a question of what instruments a healthcare setting or research lab has access to, to be able to decide what tests may be feasible for clinicians or researchers to perform. Additionally, simple molecular markers that may not require highly sensitive and quantitative results may prefer to use qPCR kits, whereas tests that require larger genome coverage or in pursuit of novel variants may require NGS.

### 3.4. CTC Detection Methods

CTCs are valuable for the prognosis and screening of metastatic cancers as low CTC count in blood is correlated with disease-free and overall survival for some metastatic cancers such as breast and colon [41]. The distinctive features of these cells allow their enrichment and detection by various methods. Filters, such as OncoQuick and Ficoll, have been used to capture CTCs based on unique physical properties, while immunological methods have been reported to utilise the expression of surface EpCAM antigens [42].

CellSearch^®^ Menarini Silicon Biosystems (Veritex, Livingston, NJ, USA) is an FDA-approved method that captures CTCs using anti-EpCAM antibodies in a magnetic ferrofluid [43]. The detection relies on the EpCAM antigen’s selective efficacy, which can add a variation in the detection of the CTC presence, which can vary across different stages of disease progression. The 2007 landmark paper for CellSearch^®^ reports an 80% recovery rate detected CTCs in approximately 70% of breast cancer patients. Additionally, the high cost (machinery cost is USD 600–800k) and lengthy sample processing times can make a wide clinical use of this test quite challenging on its current form [44]. Additionally, CellCollector^®^ is a CE-marked system utilising a structured needle with a hydrogel coating containing anti-EpCAM antibodies to bind and isolate EpCAM-positive CTCs after being inserted into patients’ veins. Unlike CellSearch, it does not require the use of expensive instruments, and the isolated CTCs can be further used for molecular analysis and phenotype characterization [45]. CellCollector^®^, when compared with CellSearch^®^, showed promise, where in vivo isolation was reported to be 58% of patients positive for at least one CTC compared with 27% for CellSearch^®^ [45]. Whilst being minimally invasive, the assay also requires a complicated enrichment step and a long detection process for the sample (30+ min). Whilst more cost-effective due to not requiring the expensive machinery, it still requires antibodies to run and so has a high run cost [44].

ClearCell FX is another CTC detection system that has been developed for the automated retrieval and enrichment of intact and viable CTCs from a 7.5 ml blood sample. Initial red blood cell lysis is necessary before enrichment, but CTCs remain in suspension, allowing for easy analysis and diagnostics downstream also due to antibody independence. Separation is performed based on the mechanical features of the CTCs, using the spiralised microchannel technology. By eliminating reliance on epithelial cell markers, the ClearCell FX system is able to capture CTCs that do not express the cell markers in addition to cells that have a mesenchymal phenotype, which would be neglected by devices like CellSearch^®^. Using the ClearCell FX system, CTCs were detected in 100% of peripheral blood samples obtained from breast cancer patients and showed a 50% recovery rate [46]. Furthermore, Parsortix^®^ (Angle PLC, Guildford, UK) has been reported as a system for capturing CTCs from liquid biopsy samples, using an instrument comprising a filtration cassette, with the captured cells to be eluted in a buffer solution for further molecular analysis [47]. They report a 81% recovery rate on whole blood spiked using cultured cell lines.

Lastly, the MagSweeper device was developed to isolate CTCs from liquid biopsy samples through immunomagnetic separation. A magnetic rod, coated in a nonadherent plastic sheath, is robotically swept through a well containing the prelabelled sample. The rod captures the labelled cells and removes them from the whole-blood sample. Through a series of washes and repeated magnetic captures, a sample of CTCs with low contamination is obtained. The CTCs were labelled in the sample using EpCAM-antibody-functionalised paramagnetic beads, which uniquely bind to the target cells, for collection by the magnetic rod. Comparative results from patients diagnosed with metastatic breast cancer and healthy controls identified CTCs in all samples derived from those with cancer and no CTCs in samples from healthy donors [48]. In 2014, Deng et al. used this technology to analyse mutations of the *PIK3CA* gene in CTCs [49]. The MagSweeper was used to capture and isolate CTCs, and although this technology was successful in obtaining viable cells for further analysis, the reliance of this technology on the presence of EpCAM antigens could lead to events where cells that have undergone EMT could remain undetected. Some examples of certified tests available for each technology type are summarised in Table 1.

### 3.5. DNA Extraction and Sample Preparation

Despite the method used for ctDNA detection, DNA extraction and sample preparation are vital steps as part of the integration of assays in a system. Sample preparation often consists of four steps: disruption of cells, protein and lipid removal, DNA purification, and then concentration of DNA.

Methods to perform these steps can vary from mechanical approaches, such as ultrasonication, which disrupt cells using pressure, to homogenisation, which shreds cells or chemical methods relying on the use of detergents that break down membranes. Premade extraction kits now exist on the market, simplifying nucleic acid extraction, with examples to include the QIAamp Circulating Nucleic Acid Kit [68] and the Zymo Quick-ccfDNA Serum and Plasma Kit [69], both of which use silica beads to bind DNA molecules, whilst the rest of the sample is washed away. The QIAamp MinElute ccfDNA Midi Kit [70] and the Maxwell RSC ccfDNA Plasma Kit [71] use magnetic beads instead to trap DNA molecules, whilst the rest of the impurities are eluted away. Whether using external extraction kits, silica-column-based precipitation methods, or centrifugation techniques that require in-house equipment, these protocols add complexity, cost, and labour to the utility of nucleic acid detection in cancer care, acting as another barrier to mutational detection for point-of-care testing.

## 4. Emerging Technologies: Moving towards a New Type of Liquid Biopsy

Currently, numerous liquid biopsy technologies exist and are all utilised in distinct ways for different cancers. NGS can offer more sensitive informative tests but has higher costs and requires high bioinformatic turnaround time in the case of WGS. Digital and nondigital PCRs can be more affordable, but still require benchtop equipment and trained staff to perform the tests. Lab-free point-of-care testing, where highly trained personnel and benchtop machines are not needed, provides a cost and personnel advantage that could allow LBs to become incorporated in clinical cancer practice, even in areas with limited equipment or staff. This section will focus on newer-generation technologies, aiming to aid LB diagnostics with cost efficiency in mind for a new era of personalised cancer management. Broadly two areas of emerging technologies are of interest: nanotechnology-based and microfluidic chip devices.

### 4.1. Nanotechnology

Nanotechnology is being used to develop new sensors and devices for the detection and analysis of liquid biopsy biomarkers, with the potential to offer greater sensitivity and specificity compared with existing technologies. One example of this is the usage of gold nanoparticles (AuNps) in capturing ctDNA during DNA hybridisation. In a study by Hu et al. (2018), gold- and silica-coated iron nanoparticles were functionalised with the oligonucleotide sequences, which are specifically complementary to the gene sequence of 7 *KRAS* point mutations [72]. When added to a liquid-biopsy-obtained solution of *KRAS* mutation containing ctDNA, both functionalised nanoparticles hybridise to the targeted mutant DNA, coupling the two nanoparticles together with the mutation-containing DNA acting as a bridge. These bridged structures can then be magnetically removed from the solution, and the concentration of gold removed can be measured by inductively coupled plasma mass spectrometry. This can indicate the frequency of *KRAS* mutation present due to the previous establishment of a linear correlation between gold and the concentration of mutations. Detection sensitivity was established, using a standard curve, as 0.1 pg/ml without prior PCR amplification in samples obtained from lung cancer patients; this is equivalent to 0.12% mutation frequency in ddPCR/NGS [72]. Further testing was performed on other mutant variants of the *KRAS* gene, where no significant difference was identified, suggesting that the technology is equally sensitive to all variants. Additionally, no gold particles were detected in solutions that did not contain the target gene, demonstrating high selectivity. Comparison of nanocoupling technology and ctDNA with ddPCR technology and tissue biopsy samples showed perfect concordance in real patient samples.

Continuing in the field of AuNp devices, recent studies have been conducted to detect *EGFR* exon 19 deletion in tumour DNA using condensed AuNp and catalytic walker DNA [73]. They used strand displacement hybridisation to continuously amplify a DNA sample whilst being immobilised on AuNps. They reported a limit of detection (LOD) of 38.5 aM for the detection of the deletion. AuNps have also been used in research to show CTC capture in addition to ctDNA detection. This year, a group was able to create a system using AuNps with a microfluidic platform to capture CTCs within 120 min at a 90% recovery rate. They compared their microfluidic system with and without antibody-coated AuNps and found that AuNps increased capture efficiency by approximately 10%, but also that they enhanced cell viability by about 20% [74], showing great promise when comparing this technology with other approved methods with lower capture rates. Further clinical testing of this platform to see how well it performs on patient samples is required for a more thorough comparison of efficacies.

### 4.2. Microfluidic-Based Devices

Microfluidic chips have the capacity to be mass-produced and fully automated, reducing complexity and increasing the affordability and accessibility of the test. Microchip design in tandem with microfluidics has become increasingly popular over the last decade. Fachin et al. (2017) developed the CTC-iChip to sort CTCs by negative selection using a combination of lateral displacement, inertial focusing, and magnetophoresis on a single plastic chip. This technology uses antibodies and magnetic beads specifically targeted to white blood cells for their removal. Meanwhile, red blood cells, platelets, plasma proteins, and other material do not fit the size threshold of the channels on the chip. The use of negative selection enables the input of whole blood into the chip, eliminating the need for excessive sample preparation. The use of negative selection also bypasses the shortcomings identified in methods reliant on CTC size and EpCAM expression, such as the CellSearch and CellCollector, as these parameters have been shown to vary significantly between cases. Operational performance was tested across 44 spiked cell line experiments with a median of 99.5% recovery rate. When using patient blood samples, a 95% recovery rate was obtained; however, after adjusting the rate for possible white blood cell contamination, an 84% recovery rate was obtained for breast cancer, 88.5% for lung patients, and 96.4% for prostate [75].

Whilst many assays have been studied to detect ctDNA, there is a lack of standardisation for optimal DNA extraction when studies attempt to miniaturise ctDNA technologies; thus, sample isolation and prep are often topics tackled by microfluidics. Solid-phase extraction microdevices, such as silica membranes or beads, can be used. Liquid-phase extraction chips can utilise electrophoresis, such as dielectrophoresis (DEP) microchips. A DEP extraction method on a chip was developed by Zhang et al., in which particles are suspended from a fluid by polarising forces. Particles are exposed to a nonuniform electric field (NUEF) to become electric dipoles. While exposed to an NUEF, the particles move in certain directions due to their polarisation and can move up and down microfluidic channels on a chip. DEP chips, therefore, separate ctDNA or CTCs based on characteristic electric signals and remain label-free, fast (<30 min), and scalable [76]. One study created a full electrical sensor lab-on-chip platform using DEP to determine the difference between posthybridised DNA and nonhybridised DNA. They tested their platform on EGFR ctDNA detection in NSCLC patients and were able to distinguish 1 fgul of a selected EGFR mutation against 100 pgul wild-type DNA, a 0.01% sensitivity [77].

Furthermore, the inherent advantages of affordability and scalability that semiconductor technology brings are ideal for the development of the next generation of healthcare devices, where miniaturisation, low cost of fabrication, intelligence in processing and AI integration, high-throughput screening, and reusability are all factored in the design of a detection platform. Specifically, the combination of CMOS (complementary metal–oxide–semiconductor) microchip technology and ISFETs (ion-sensitive field-effect transistors) results in silicon-integrated sensors. These can be coupled with microchip-compatible amplification methods for the detection of cancer-specific targets, which has been successfully demonstrated in recent case studies. For more than a decade, novel CMOS-integrated ISFET-based platforms in the form of lab-on-chip systems have been fabricated and tested, allowing for the label-free detection of molecular targets, following the principle of hydrogen ion release when amplification occurs in the presence of a specific target. The original concept of detecting DNA using an ISFET was introduced by Toumazou et al. [78], demonstrating the ability of the sensor to detect the presence of a nucleic acid and detect it as DNA amplification occurs, where DNA bases are incorporated into an elongating DNA strand in the presence of dNTPs and a DNA polymerase. This was the foundation for ISFETs to be later applied as the sensing elements of the next generation of large-scale semiconductor sequencing platforms [79], unravelling their potential as integrated sensors in the fields of diagnostics.

Research in ISFET sensing has thrived with continued exploration across various fields, combining its hydrogen ion sensing capabilities with compatible amplification methods, such as loop-mediated isothermal amplification (LAMP) [80]. LAMP operates in isothermal conditions, producing a higher yield of amplification compared with standard PCR-based methods. It has been applied in numerous studies within the field of rapid diagnostics, especially when coupled with ISFET-based nucleic acid detection, resulting in a more robust pH signal monitored by the sensors. This is generated through sequence-specific amplification reactions, all achievable without the need for a thermal cycler, thus allowing the chemistry to function within a miniaturised lab-on-chip device. Demonstration of this work has been shown in a range of applications in the field of diagnostics [81,82,83,84] and particularly in cancer, with the latest features of its capability to be presented for the detection of clinically validated cancer-specific mutations in breast cancer (*ESR1*, *PIK3CA*) [85,86,87], circulating mRNA biomarkers in prostate cancer (TMPRSS2-ERG, YAP1, AR-V7) [88], tumour-specific markers (HPV-16/18, hTERT mRNA) in cervical cancer [89], and DNA methylation biomarkers present in several cancer types [90,91,92]. A summary of all the emerging technologies that have been discussed can be found in Table 2 and an image illustrating these technologies can be seen in Figure 2.

## 5. Conclusions

Liquid biopsy technologies have great potential to facilitate the integration of precision oncology into routine clinical practice, thus aiding diagnosis and treatment stratification through the use of technological platforms that will allow longitudinal monitoring of tumour growth and evolution across a patient’s diagnosis and treatment timeline. Accessible and tumour-informed diagnostic platforms are anticipated to have major patient benefits on treatment selection and early- and advanced-stage disease monitoring. There are numerous examples of existing LB-based technologies and platforms as these have been summarised in this review, highlighting the academic progress and innovation for the next generation of liquid biopsy assays and technological platforms. Another important aspect is the correlation between the molecular characterisation and understanding of tumour mutations and their link to bespoke therapeutics, leading to precision in cancer care. Not all cancers currently lead to clinical actions taken with information from the genetic makeup of the patient’s tumour taken into consideration. Consequently, the utility of nucleic acid detection in cancer care is limited by the information that it brings, which can then inform towards further actions taken.

Whilst ctDNA and CTC detection are still not being heavily used as part of a cancer patient management diagnostic and monitoring plan, FDA approvals for assays based on PCR, ddPCR, and NGS are gaining ground. Specifically targeted NGS panels are bridging the gap by reducing the cost and complexity of NGS to be more suitable for healthcare whilst still providing multiple gene targets per assay. For all assays and technologies in this space of LBs, they have to compete over factors such as sensitivity, amount of input ctDNA, fragment detection, and specificity. These factors are causing the field to constantly evolve, whether that involves older methods becoming more capable and accessible like targeted panels using NGS, or the emergence of a new technology like biosensing-capable chips. Nonetheless, affordable and reliable cancer diagnostic devices and tests are likely to make widespread genetic testing a reality.

## Figures and Tables

**Figure 1 cancers-15-05434-f001:**
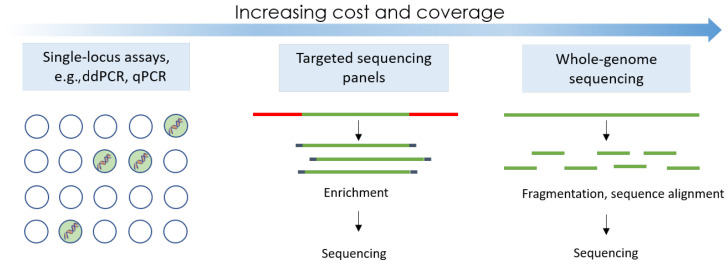
The three main technological principles used in the majority of liquid biopsy assays.

**Figure 2 cancers-15-05434-f002:**
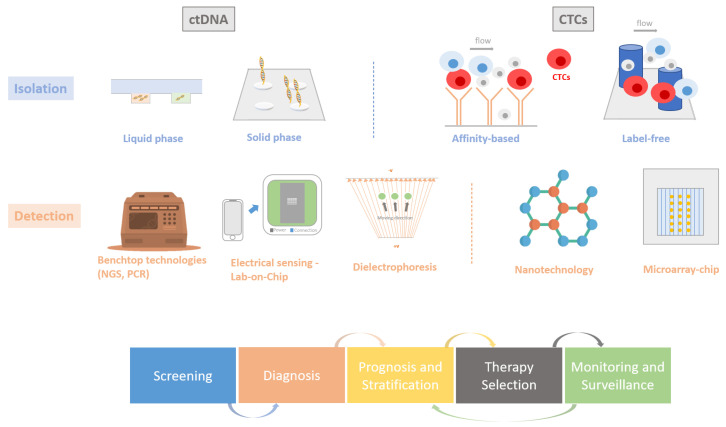
Various methods for ctDNA and CTCs isolation and mutational detection. The third row indicate the logical workflow of patient care for patients with cancer, where the arrows indicate the next step that can be performed.

**Table 1 cancers-15-05434-t001:** Summary of key selected liquid biopsy tests and technological platforms presented for the detection of ctDNA and CTCs.

Testing System	Cancer Type	Technology	Ref.
ctDNA Assays			
Cobas EGFR v2	Non-small cell lung cancer (NSCLC)	Nondigital, PCR kit	[3]
CancerSEEK	Various	qPCR	[22]
Therascreen	NSCLC, breast cancer	qPCR	[24]
Guardant 360	Colorectal, breast, NSCLC	NGS	[50]
FoundationOne Liquid CDx	Various	NGS	[37,38]
GRAIL	Various	NGS	[51]
OncoBEAM	NSCLC, colorectal, melanoma	BEAMing (ddPCR)	[52]
Precipio	NSCLC	Ice-cold PCR	[53]
Freenome	Colorectal, prostate	Multiomics	[54]
Oncomine	Lung, breast, and others	NGS	[55]
Signatera	Various	NGS	[56]
Idylla	Lung, colorectal	PCR	[57]
Sysmex Safe-Seq	Breast cancer, head and neck cancer	NGS	[58]
CTC Assays			
Cell Search	Breast, prostate, colorectal	Ferrofluid nanoparticles and antibody	[43,59]
ClearCell FX1 System	Breast, lung	DFF, microfluidics	[46,60]
GILUPI Cell Collector	Lung, colorectal	Anti-EpCam antibodies	[45,61]
AccuCyte ®, CyteFinder ®	Prostate, breast, lung	Density-based separation, imaging	[62,63]
Parsortix	Various	Microfluidics	[47]
OncoQuick; Ficoll	Gastrointestinal, colorectal	Density gradient centrifugation	[64]
MagSweeper	Breast, colorectal	Antibodies	[65,66]
ImageStream	Hepatocellular carcinoma	Flow cytometry and immunofluorescence	[67]

**Table 2 cancers-15-05434-t002:** Emerging ctDNA and CTC technologies for liquid biopsies, alongside the cell lines on which they were tested.

Name	Technology Description	Cell Line(s)	Ref.
ctDNA Assays			
Lab-on-Chip	ISFET-enabled CMOS microchip	Breast cancer	[85,86]
Huang Biosensor	Isothermal nest hybridisation of DNA for activation of HCR products that generate a quantifiable electrochemical signal	Breast cancer	[93]
Rahman Biosensor	Single-stranded DNA probes immobilised onto a nanoplatform	Gastric cancer	[94]
Zhang Electrochemical Sensor	Single-stranded DNA probes bound to Mo2-containing nanosheets	Gastric cancer	[95]
Hu Nanoparticle Sensor	Mutation-specific functionalised iron nanoparticles coated in gold and silica, specifically complementary to target DNS	Colorectal cancer	[72]
CTC Assays			
CTC-Chip	EpCAM-antibody-coated microchip	NSCLC and other	[96]
NanoVelcro-Chip	EpCAM-antibody-coated silicon nanowires	Prostate cancer	[97]
Herringbone-Chip	EpCAM-antibody-coated, ridged microchannels	Prostate cancer	[98]
EasySep CTC Enrichment Kit	Negative selection via CD45 markers	Gallbladder cancer	[99]
RosetteSep CTC Enrichment Kit	Antibody-mediated cross-linkage and negative selection of undesirable cells	Prostate cancer	[100]
CTC-iChip	CTC isolation by lateral displacement, inertial focusing, and magnetophoresis		[75]
Warkaini’s Chip	Physical separation of CTCs using silicone spiralised microchannels	NSCLC and breast cancer	[101]

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
