# Peer review of "The Evolution of Affordable Technologies in Liquid Biopsy Diagnostics: The Key to Clinical Implementation"

_cancers, 2023, doi:10.3390/cancers15225434_

Round 1

Reviewer 1 Report

Comments and Suggestions for Authors

 The reviw by Alexandrou et al. aims to provide an overview on the technologies currently available for the use of liquid biopsy in diagnosis. To this purpose, besides   of describing the most widely approaches presently adopted for the analyses of ctDNA and CTCs, they also focused on the promising introduction of nanotechnologies.

1) In order to provide a substantial comparison, I would recommend to introduce a new Table summurizing the different sensibility/specificity of all methods (e.g limit of detection), including the nanotechnologies approaches, together with their costs, divided specifically for ctDNA and CTCs.

2) The authos have underlined the importance of DNA extraction for the novel nanotechnologies, however, this point is relevant also for PCR/NGS methods, particularly when clinical applications are implied. It is well known that tumors can release differently according to the type and phase of the disease. A paragraph focused more specifically on the state-of - art of the DNA extractionwould be needed. This point can dramatically influence the following analytical assays.

3) Minor points: the sentence of line 119 should be edited, whereas the sentence of line 187 is not clear.

Comments on the Quality of English Language

English language is fine, just two lines to be revised: lines 119 and 187

Author Response

Please see the attatchment.

Reviewer 2 Report

Comments and Suggestions for Authors

In their review, Alexandrou and colleagues adeptly illuminate the latest technologies in liquid biopsy and delve into various aspects of clinical cancer management. Overall, the review exhibits a well-structured format, providing essential information and references. However, addressing minor concerns could further enhance the manuscript.

Given that the authors present their perspective, citing their own work seems acceptable. Nevertheless, the conclusions might benefit from a more explicit acknowledgment of the limitations within this rapidly evolving field. While screening or monitoring may prove beneficial for certain tumors, it may offer limited utility for those with no available treatment options, such as glioblastoma. Additionally, the cost-effectiveness of these approaches appears somewhat ambiguous.

Minor Issues:

  1. Line 22: While the numerical values may remain relatively stable, considering the possibility of an outdated reference or incorporating a more current one could enhance the credibility of the information.

  2. Lines 24-25: Including references for the mentioned FDA test and additional tests would strengthen the validity of the information presented.

  3. Line 26: Although not mandatory, providing information on the leading causes of death could offer valuable context to the reader.

  4. Line 91: It's worth noting that circulating tumor cells (CTCs) can originate not only from the primary tumor but also from metastatic sites.

  5. Lines 95-97: In addition to immune system factors, mechanical forces such as shear stress may contribute to the elimination of circulating cells. Acknowledging these mechanical factors would provide a more comprehensive understanding of the dynamics involved.

Reviewer 3 Report

Comments and Suggestions for Authors

The review by Alexandrou et al. is overall reasonably well written but could clearly be improved. This is regarding English language but also logic and well written presentation as well as inclusion of details that would make the review relevant to readers. The review explores the changes in technologies over the last 15 or so years to analyse liquid biopsies and the move of their utility into clinical settings.  Overall, this is a reasonably comprehensive review of technologies, however, mostly lacking detailed information/references how these technologies have been applied successfully on liquid biopsies. Also, for some of the referencing that is provided key studies are not included. There are quite a few issues which the authors should address.

Issues:

1.      1. Title: replace "widespread adoption" with something like: "clinical implementation"

2.      Line 3: what do you mean with "to curve"?

3.      Line 7, “samples”: do you mean blood?

4.      Line 24: provide detail what the test was and a reference

5.      Line 27: what is "even with the rise of LBs" supposed to mean?

6.      While not disputing that the impact of COVID on healthcare is interesting I do not believe the inclusion of a long section (line 32-48) is relevant for this review/topic. This section should be deleted.

7.      Line 50: acronym “ctDNA” not introduced

8.      Line 76: “patients’ journey” replace with ”disease”

9.      Line 83: provide a reference for “actively released”

10.   Line 98: sounds as if cells become CTCs and afterwards undergo EMT. One would assume that EMT already happened before the cells are in the blood stream and referred to as CTCs. In fact that EMT allowed them to do so.

11.   Line 102: clarify response to what

12.   Most of the technology sections remain very superficial and purely descriptive rather than going into detail and examples. I suggest a reader would want to know which technology may be better and in what circumstances and the guidance (if any) the authors provide is not backed up. For instance, for ddPCR (but this applies to other technologies as well) the authors should go through a few examples of what clinically relevant biomarkers have been detected in liquid biopsy (vs tissue if applicable) with what sensitivity and specificity in important studies using that method and include the relevant citations for those studies.

13.   Line 116, change “new” to “other”

14.   Line 122,  ”prognosis”, do you mean diagnosis?

15.   Figure 1: replace “The three main detection principles used for current liquid biopsy assays: single locus assays, targeted panels and whole genome sequencing. ddPCR (digital PCR)” with a short title and add a legend to this figure.

16.   Line 130: you say dPCR in Figure 1 and ddPCR here, should be consistent.

17.   Line 146, “this oncogenic pathway: what oncogenic pathway? Clarify

18.   Line 157-158, “has a higher risk for contamination: what do you mean? Is more sensitive and thus more likely to detect also possible, very low concentrations of contaminants? Certainly, all LB molecular techniques require very clean uncontaminated conditions and clear pre and post PCR separations.

19.   Line 158-159, “have proper training before performing the procedure is silly to say, every molecular technique requires proper training to perform.

20.   ddPCR and specifically NGS sections provide company names for readers to be able to access those technologies

21.   Line 185, “?”: provide reference

22.   Line 205: add “add anti-“ before “EpCAM”

23.   Line 206: what antigen?

24.   Line 214-215: sentence does not make sense.

25.   Table 1: what does “latest” mean? brought onto the market in which years? the last 5? 10? 15? Are you aiming to be complete and inclusive? It seems to be selective. References seem to go back as far as 2007 and 2010 for CTCs with a lot of instruments, chips missing. For ctDNA assays the omission of ddPCR other than Beaming is obvious despite that technology being the likely most sensitive and widely used method to detect biomarkers in cfDNA and LB more broadly (including CTCs). At the very least change the title to “Selective liquid biopsy tests and technological platforms used for detection of ctDNA and CTCs”

26.   Line 246: given that you mainly talk about targeted NGS, turnaround time and bioinformatics involved are not that demanding.

27.   Line 259: replace “during”

28.   Line 262: which KRAS mutation?

29.   Line 269, “sensitivity was established, using a standard curve, as 0.1 pg/ml: per mL what? blood? plasma? Also, provide copies rather than pg (for your own good it is worth to calculate this assuming 170bp length of ctDNA. 0.1pg = 97.8x10^6 base pairs! 97.8x10^6 /170= 570000 copies if I calculated this correct. Meaning,  if 0.1 pg  is really the sensitivity of this assay (ie 570000 copies are needed for detection) it is extremely poor in comparison to previously mentioned methods (it is worth to review this and get this right!).

30.   Line 280-281, 3.85 x 107 nM of target DNA: again should convert into copies to allow comparison with other methods.

31.   Line 281, “a remarkably low LOD: really? You need to convince the reader: compare to other tests mentioned in the review in a table with a common measure either convert all to copy numbers  or all to nM of target DNA.

32.   Line 281, “although not fully reproducible: what does this mean? Clarify!

33.   Line 284: to capture CTCs within 120 min at 90% efficiency: how does that compare to other methods? more details needed

34.   Line 295, “by lateral displacement: explain what that means

35.   Line 300: replace chemistries

36.   Line 304, dielectrophoresis (DEP) microchips needs detail: what are they? What can they do?

37.   Figure 2: generate shorter title

Comments on the Quality of English Language

as indicated

Round 2

Reviewer 1 Report

Comments and Suggestions for Authors

The authors have  implemented the previous version of the manuscript according to the reviewer's suggestions.

In the present form the manuscript provides a novel complete overview of the main features of the different methods presently available for the liquid biopsy use.

Reviewer 3 Report

Comments and Suggestions for Authors

The authors have addressed the previous issues in satisfactory fashion.

Comments on the Quality of English Language

As ticked above.